# Polyphenol Effects on Splenic Cytokine Response in Post-Weaning Contactin 1-Overexpressing Transgenic Mice

**DOI:** 10.3390/molecules24122205

**Published:** 2019-06-12

**Authors:** Thea Magrone, Anna Spagnoletta, Antonella Bizzoca, Matteo Antonio Russo, Emilio Jirillo, Gianfranco Gennarini

**Affiliations:** 1Department of Basic Medical Sciences, Neurosciences and Sensory Organs, School of Medicine, University of Bari, 70124 Bari, Italy; anna.spagnoletta@email.it (A.S.); antonella.bizzoca@uniba.it (A.B.); emilio.jirillo@uniba.it (E.J.); gianfranco.gennarini@uniba.it (G.G.); 2MEBIC Consortium, San Raffaele Open University of Rome and IRCCS San Raffaele Pisana of Rome, 00166 Rome, Italy; matteoantonio.russo@uniroma1.it

**Keywords:** cellular and molecular rehabilitation, contactins, cytokine, neonatal immunity, polyphenols, splenocytes

## Abstract

Background: In mice, postnatal immune development has previously been investigated, and evidence of a delayed maturation of the adaptive immune response has been detected. Methods: In this study, the effects of red grape polyphenol oral administration on the murine immune response were explored using pregnant mice (TAG/F3 transgenic and wild type (wt) mice) as the animal model. The study was performed during pregnancy as well as during lactation until postnatal day 8. Suckling pups from polyphenol-administered dams as well as day 30 post-weaning pups (dietary-administered with polyphenols) were used. Polyphenol effects were evaluated, measuring splenic cytokine secretion. Results: Phorbol myristate acetate-activated splenocytes underwent the highest cytokine production at day 30 in both wt and TAG/F3 mice. In the latter, release of interferon (IFN)-γ and tumor necrosis factor (TNF)-α was found to be higher than in the wt counterpart. In this context, polyphenols exerted modulating activities on day 30 TAG/F3 mice, inducing release of interleukin (IL)-10 in hetero mice while abrogating release of IL-2, IFN-γ, TNF-α, IL-6, and IL-4 in homo and hetero mice. Conclusion: Polyphenols are able to prevent the development of an inflammatory/allergic profile in postnatal TAG/F3 mice.

## 1. Introduction

In mice, the post-natal immune system is not completely developed, and maternal milk seems to play an important yet protective function [1]. Therefore, evaluation of the murine immune system function during lactation and after weaning may provide a wealth of useful information for a better understanding of its developmental evolution and regulation.

Within the lymphoid tissue, the thymus represents the major site for T lymphocyte maturation [2], which occurs close to birth and involves both CD4+ T helper (h) and cytotoxic CD8+ T cells, which become more responsive to mitogens compared to other immune cells, as deduced by exploring their proliferation in mixed cell cultures [3]. Thymic maturation progresses postnatally, as evidenced by the robust T cell response to concanavalin A, which occurs between 2 and 3 weeks of age [4].

As for the spleen, by the end of the gestation, this organ is still immature from an immunological point of view, but by 3–7 days postnatally, the splenic periarteriolar lymphocyte sheet (mainly composed by T cells) starts developing, with the marginal sinus becoming more evident by day 7 [5]. Primary follicles appear 7–14 days postnatally, when they represent the main site of splenic B lymphocytes and of follicular dendritic cell concentration [5]. As for lymph node development, mesenteric lymph nodes first appear mid-gestation, while the popliteal ones become evident close to birth [6,7]. At this stage, lymph nodes are still immature, T cell colonization and formation of primary follicles occurring by the end of the first postnatal week and their germinal centers appearing at week 4 [8,9].

The development of the murine adaptive immunity, which occurs in the immediate postnatal period, has been addressed in a few studies. In neonatal mice, reduced T cell number has been demonstrated, when compared to adult mice [10,11]. This is functionally consistent with the downregulation of serum interferon (IFN)-γ levels in response to toll-like receptor (TLR) stimulation [12]. These data indicate that Th1 cell function, as an expression of adaptive immunity, is impaired in neonatal mice, thus leading to expanded innate immune responsiveness. Conversely, adult murine T cells were found to downmodulate uncontrolled inflammatory responses, which may be harmful to the host [13]. In support of previous concepts, other murine studies documented that early CD4+ thymic emigrants are rather Th2 cells and produce interleukins (IL)-4, IL-5, and IL-13, respectively, with a lower polarization towards the Th1 subset [14]. However, this event may lead to loss of protective activity against pathogens. In this framework, a further report evidenced that the murine Th2 locus undergoes epigenetic modulations during fetal thymic ontogenesis until the first week of postnatal life [15].

Taken together, these data point out a certain degree of vulnerability of neonatal mice to infections, which may result from impaired cell-mediated adaptive immune response imparted by Th1 cells. To overcome delayed murine Th1 cell maturation, dietary manipulations with prebiotics, galacto-oligosaccharides, fructo-oligosaccharides, fish oil, and synbiotics have been demonstrated to be very effective [16,17,18,19,20,21,22].

Polyphenols (flavonoid and nonflavonoid compounds) are largely present in fruits, vegetables, cereals, wine, and extra virgin olive oil [23]. These natural substances assumed with foods are able to modulate intestinal and systemic immune responses [24]. Particularly, red wine- and red grape-derived-polyphenols display a plethora of anti-inflammatory and anti-allergic activities. Indeed, they compete with endotoxin binding to human TLR-4, thus inhibiting activation of the nuclear factor kappa-light-chain-enhancer of the activated B cells pathway and pro-inflammatory cytokine secretion [25]. In mice, administration of polyphenols from fermented grape marc (FGM) attenuated experimental colitis [26] as well as asthma [27]. In in vitro experiments, FGM induced activation of T regulatory (Treg) cells in human healthy peripheral T lymphocytes and inhibited human basophil degranulation [28].

Taking advantage of the abovementioned polyphenol activities, in this study, splenic immune responsiveness was evaluated in suckling pups from mothers orally administered with polyphenols between 8 and 30 days after weaning. TAG/F3 mice, a transgenic line undergoing delayed neural development as a consequence of upregulation of the Contactin 1 adhesive glycoprotein expression, were employed [29,30,31,32,33]. Of note, in a companion paper, the effects of polyphenol administration were evaluated on cerebellar neurogenesis in postnatal TAG/F3 pups (manuscript under review). Here, we report that a weak immune response in pups on day 8 is evident, while a more vigorous immune activation is found on day 30. Some differences are also detectable between wild type (wt) and TAG/F3 mutant mice following phorbol myristate acetate (PMA) stimulation in terms of Th1/Th2 functions with a tendency to a more pronounced Th1 and Th2 cytokine response in the latter. On the other hand, polyphenol administration in pups after weaning seems to mitigate the enhanced cytokine production, thus preventing potential development of inflammatory/allergic diseases.

## 2. Results

### 2.1. Th1-Related Cytokines

IL-12 is a Th1-induced cytokine, released by splenic dendritic cells upon antigenic activation [34]. As shown in Figure 1, at post-natal day 8, IL-12 could not be detected in supernatants from wt splenocytes in the absence of treatment or in mice undergoing PMA stimulation and in suckling pups which received milk from dams administered with polyphenols (POP mice). Similarly, no IL-12 was detected in wt POP mice undergoing PMA stimulation or in homo TAG/F3 transgenic mice (+/+) in the absence of any treatment. IL-12 was still undetectable in supernatants from TAG/F3 murine splenocytes in the presence of polyphenol administration (+/+/POP) as well as from hetero mice (+/−/POP) undergoing PMA administration. IL-12 was also undetectable in supernatants from the homo POP and POP/PMA mice as well as in the hetero POP and hetero PMA mice. On the other hand, high IL-12 levels were found in supernatants from homo TAG/F3 mice as well as in hetero POP mice, upon PMA treatment, respectively. The latter resulted in a marked IL-12 release higher than that observed in homo TAG/F3 mice.

IL-2 is a major growth factor released by activated Th1 cells [35]. At postnatal day 8, it was absent from both control and active supernatants in all mice (data not shown). However, a differential IL-2 expression was observed at postnatal day 30 upon PMA stimulation, which gave rise to IL-2 release in both wt and wt post-weaning mice that received conventional feed supplemented with polyphenols (POMI) (Figure 2). In homo TAG/F3 murine splenocytes, background levels of IL-2 were still observed, which, however, were markedly increased upon PMA stimulation, although the corresponding values remained lower than those observed in PMA-stimulated wt mice. Similarly to wt mice, no IL-2 release was observed in both homo POMI and homo POMI/PMA and in hetero mice in the absence of PMA stimulation. By contrast, in hetero mice, PMA treatment induced a sharp IL-2 increase from the background levels. Although to a lower extent, such a PMA-mediated effect was also observed in the hetero POMI murine strains, in which PMA-activated splenocytes released higher amounts of IL-2 compared to those observed in the untreated hetero POMI counterpart.

IFN-γ is a predominantly Th1-derived cytokine, which plays a fundamental role in host protection against intracellular pathogens [36].

At postnatal day 8, this cytokine could not be detected in control and active supernatants from both wt homo and hetero pups, respectively (data not shown). At postnatal day 30 (Figure 3), minute amounts of IFN-γ were detected in wt mice supernatants upon PMA stimulation. In wt POMI/PMA mice, tiny amounts of IFN-γ were detected. However, in both homo and hetero mice, a very sharp effect in terms of IFN-γ production was observed upon PMA treatment. These effects were abrogated by polyphenol administration per se or in association with PMA.

### 2.2. Th2-Related Cytokines

IL-4 was identified as the major cytokine produced by Th2 cells [37]. The corresponding results are reported in Figure 4. At day 8, IL-4 was not detected in the presence or absence of polyphenols (data not shown). At day 30, IL-4 was present in tiny amounts in wt mice upon PMA stimulation, while POMI mice failed to induce release of this cytokine. No IL-4 was detected in homo TAG/F3 mice in all experimental conditions. By contrast, in hetero TAG/F3 mice, PMA activation led to a marked release of IL-4 in the absence of polyphenol administration.

IL-10 is an anti-inflammatory cytokine released by Treg cells [38]. As shown in Figure 5, at day 8, IL-10 release was not detected in wt pups. Conversely, in both homo and hetero TAG/F3 mice, PMA stimulation generated elevated amounts of IL-10 (higher in homo than in hetero mice). In hetero POP mice, IL-10 could be detected at a concentration higher than that observed in the control PMA-activated counterpart. Polyphenol treatment of POP mice per se failed to induce release of IL-10 irrespective of the genotype.

At day 30, no evidence of IL-10 release was recorded in all experimental setups (data not shown).

At days 8 and 30, IL-17, an inflammatory cytokine produced by Th17 cells [39], was not detected in wt, homo, and hetero mice (data not shown).

IL-6 is an acute phase protein considered as an expression of innate immune response [40].

At day 8, IL-6 was not detected in all pups (data not shown). As indicated in Figure 6, even if at a different magnitude (wt > hetero > homo), only PMA-stimulated splenocytes were able to generate IL-6 at day 30.

TNF-α is another cytokine mostly produced by macrophages and implicated in chronic inflammatory disease [41]. Data related to TNF-α release are reported in Figure 7. At day 8 (Figure 7A), POP wt mice, in the absence or presence of PMA, released a remarkable amount of TNF-α (higher in POP mice than in POP/PMA animals). In TAG/F3 homo mice, PMA stimulation resulted in TNF-α release, which, however, was lower than that seen in the wt counterpart. No TNF-α release was noted in hetero mice. At day 30 (Figure 7B), in control PMA-stimulated wt mice, TNF-α was produced at levels, which were higher than those observed in the PMA-stimulated splenocytes from wt POMI mice. In both homo and hetero mice, PMA stimulation only resulted in TNF-α production, which was higher in homo mice than in the hetero counterpart.

## 3. Discussion

The present data confirm and extend previous attainments on murine neonatal immune system maturation [4,5]. Indeed, data show that in both wt and mutant mice undergoing misexpression of these axonal adhesive glycoproteins, PMA stimulation generates release of splenic cytokines, which are preferentially delivered at postnatal day 30 compared to day 8. The time of appearance and the concentration profiles of various released cytokines were compared between wt and TAG/F3 transgenic mice.

To the best of our knowledge, this is the first observation on immune system development in TAG/F3 transgenic mice. In the present discussion, the main arising evidence is that at days 8 and 30 in both wt and TAG/F3 transgenic mice, unstimulated cells are unable to release any type of cytokine. Therefore, only data related to PMA-stimulated cells with or without polyphenol administration were analyzed. At postnatal day 8, IL-12, a cytokine produced by antigen presenting cells (e.g., splenic dendritic cells and macrophages) [34] is detectable neither in wt nor in hetero mice. On the other hand, functionally elevated amounts of such a cytokine are produced by activated splenic cells from TAG/F3 homo mice. By contrast, at day 30, IL-12 is undetectable in all mice. Since IL-12 polarizes the adaptive immune response toward the Th1 cell phenotype [34], this study has also been focusing on the release of related cytokines, including IL-2 and IFN-γ. IL-2, a powerful growth factor for both T and B lymphocytes [35], is secreted only at day 30 and, upon PMA stimulation, it can be detected in wt (50 ± 1.92 pg/mL), homo (33 ± 2.06 pg/mL) and hetero TAG/F3 mice (59 ± 0.82 pg/mL), thus suggesting that lymphocyte proliferation may have started.

As far as IFN-γ is concerned, at day 8, this cytokine is undetectable in all mice, while at day 30, it is produced in higher amounts in both homo (35 ± 0.95 pg/mL) and hetero mice (31 ± 0.82 pg/mL) when compared to the wt counterpart (7 ± 0.82 pg/mL). IFN-γ is a protective cytokine which promotes the microbicidal function of macrophages as well as cytotoxic functions of T lymphocytes (CD8+ cells) and natural killer (NK) cells which, in turn, exert anti-neoplastic (CD8+ and NK cells) and antiviral functions (CD8+ cells), respectively [36]. All together, these data suggest that Contactin 1-overexpressing transgenic mice should be more protected against pathogens in postnatal life compared to wt mice.

With regard to IL-4 a Th2-related cytokine [37], at day 8, there is no evidence of its release. IL-4 rather appears in tiny amounts (2 ± 1.15 pg/mL) at day 30 in wt mice, although it is more abundant in hetero mice (60 ± 1.82 pg/mL). On the other hand, homo mice lose this function.

Of note, IL-4 is involved in allergic response (IgE production) and in the conversion of the primary (IgM) to the secondary (IgG) immune response [37].

IL-10 is a cytokine predominantly produced by Treg cells, provided with anti-inflammatory and immune modulating activities [38]. In wt mice, no evidence of IL-10 release was present at days 8 and 30. Conversely, it was greatly produced at day 8 in both homo (138 ± 0.82 pg/mL) and hetero (75 ± 1.14 pg/mL) mice. Although the interpretation of this set of data may be controversial, one can hypothesize that early production of IL-10 in transgenic mice may represent the residual expression of its fetal release, aimed at contrasting the maternal immune response against the embryo. In support of this hypothesis, in 30-day-old transgenic mice, IL-10 production is no longer evident.

IL-17, an inflammatory cytokine predominantly released by Th17 cells [39], is undetectable in all mice at days 8 and 30. Then, at this time of the postnatal immune system development, the IL-10/IL-17 equilibrium is not evident at least at splenic level. This fact may depend upon murine gut microbiota development, since intestinal bacteria (the so-called microbiota) are able to influence the IL-10/IL-17 ratio [42]. To the best of our knowledge, no data are available on the composition of intestinal microbiota in the mice used in this study.

IL-6 is an acute phase protein also involved in Th17 cell activation [40]. Its release is undetectable in all mice at day 8, while it is weakly expressed at day 30 in homo (1.25 ± 0.95 pg/mL), hetero (6 ± 0.81 pg/mL), and wt mice (6.75 ± 0.95 pg/mL), respectively.

TNF-α is an inflammatory cytokine, which actively participates to the maintenance of chronic and auto-immune disease [41]. At day 8, no TNF-α release was found in wt and hetero mice, while it was weakly present in homo mice (6 ± 0.82 pg/mL). At day 30, TNF-α production was quite abundant in wt (64.5 ± 1.71 pg/mL), homo (115 ± 0.82 pg/mL), and hetero (41.7 ± 0.96 pg/mL) mice. This indicates that splenic macrophages may represent a great source of TNF-α in response to environmental antigenic stimuli. As discussed above, IFN-γ-induced potentiation of macrophage function may also lead to secretion of TNF-α by inflammatory macrophages (M1 type) [36].

In synthesis, all mice seem to be protected against external and internal stimuli, even if the absence of IL-10 production may result in exaggerated inflammatory responses. These data also suggest a powerful response of the innate immunity at early stages.

Previous studies on TAG/F3 transgenic mice [29,32,43] demonstrated a consistent developmental delay in different neural structures of the arising transgenic mice, occurring in the cerebellar [29] and the cerebral [32] cortices as well as in the hippocampus [43] and the basal ganglia [44]. In terms of the underlying mechanisms, the observed phenotype implies the activation of developmental control genes. Indeed, the activation of specific signaling pathways and, in particular, those involving the Notch receptors [30,31,32] and the phosphorylated cAMP response element-binding protein transcription factor has been reported [45]. However, the putative relationship between alterations of the nervous system and the observed immune profile in TAG/F3 mice deserves further studies also in view of the notion that Contactins are expressed in the nervous system only.

As previously stated in this paper, polyphenols are endowed with anti-inflammatory and immune modulating activities in various experimental and clinical settings [23,24,25,26,27,28]. On these grounds, the impact of polyphenols on postnatal development of the murine immune system was investigated. From the overall data, it emerged that these substances exert a strong influence on murine PMA-activated splenic cells. At day 30, in wt mice, there was evidence of a polyphenol-mediated potentiation of IL-2 release (56 ± 1.82 pg/mL). Furthermore, at day 8 in polyphenol-administered wt mice, in the presence or absence of PMA, TNF-α release can be detected (26 ± 0.81 pg/mL and 36 ± 1.82 pg/mL, respectively), and its production still persisted up to day 30. One can hypothesize that in wt mice, this effect may rely upon the observed lack of IL-10 production despite polyphenol absorption (day 8) or its administration (day 30). Conversely, in hetero mice, polyphenols exert a marked effect both at day 8, inducing IL-12 (38.75 ± 14.84 pg/mL), and at day 30, inducing IL-10 release (88.25 ± 1.26 pg/mL). On the other hand, under the influence of polyphenols, PMA-induced increase in IFN-γ, IL-4, IL-6, and TNF-α in homo and hetero POMI is abrogated. These data may suggest that Contactin-overexpressing transgenic mice undergo more rapid immune maturation, especially in the homo mice compared to the wt counterpart and that polyphenols tend to temperate polarization of the immune response toward inflammation. In support of this contention, polyphenol-modulating activity appears to be more marked in polyphenol-untreated hetero mothers at 180 days when these mice release elevated levels of IL-2, IFN-γ, IL-4, IL-6, TNF-α, and IL-17 in response to PMA stimulation (unpublished observations). In the same mice, such an immune inflammatory profile is dramatically reduced by polyphenol administration.

In Figure 8, the development of the cytokine response in spleens of wt and TAG/F3 mice and the effects of polyphenols are summarized.

## 4. Materials and Methods

### 4.1. Mice Strains

The generation of the TAG/F3 transgenic mice strain, which undergoes delayed neural development as a consequence of Contactin 1 and subsequent Notch pathway upregulation, has already been described [32,33]. Mice were allowed free access to food and water, housed at constant room temperature (20–22 °C) and exposed to a light/dark cycle of 12 h/day (08.00–20.00 h).

### 4.2. Polyphenol Source and Addition to Feed

*Canosina* red grape from Nero di Troia is an autochthonous *Vitis vinifera* grape cultivar, growing in Apulia (South Italy), characterized by thick-skinned and small-sized berries. Frozen seeds from berries were extracted by percolation with ethanol/water (70:30). The extract was first analyzed by liquid chromatography with diode array detection to define the polyphenol composition. Thereafter, it was purified on a synthetic adsorbent brominated resin, and the polyphenol content percentage was determined. Extracts were evaluated for their potential antioxidant activity using the 2,2-diphenyl-1-picrylhydrazyl assay, which measures the ability of test agents to scavenge radicals [46].

### 4.3. Experimental Design

The following groups of mice received polyphenols supplemented with 92.5 µg grape seed polyphenols/gr and conventional feed (Harlan 2018):

#### Dams during Pregnancy and Lactation

1. Post-weaning mice (day 21 onward). For simplicity, these mice are indicated as POMI in the text and in the figures. They ingested 32 to 92.5 µg of polyphenols/day up to day 30.

2. Suckling pups. Up to day 21, suckling pups received milk from dams administered with polyphenols. These mice are indicated in the text and figures as POP. In the context of the present study, six groups of postnatal mice were processed, which included wt mice, as well as mice carrying the TAG/F3 mutation in homozygosis (homo) and heretozygosis (hetero). For these groups, both polyphenol-treated and -untreated mice were evaluated, respectively.

Animals were bred at the Department of Basic Medical Sciences, Neuroscience and Sensory Organs, University of Bari, Bari (Italy), and experimentation conformed to the EU directive 2010/63/EU by following the Italian Ministry of Health law of March 4, 2014, n. 26 upon the Authorization n. 982/2016 released from the Italian Ministry of Health on October 17th 2016, which also includes the criteria for the ethical approval of the proposed research. The “Organism in charge of animal welfare” of the Bari University conformed to the above rules.

Before sample collection, mice were anaesthetized by an intraperitoneal injection of Avertin (2,2,2-tribromoethanol) 0.45 µg/g body weight followed by cervical dislocation.

Specimens were taken from spleen of at least 5 mice from each genotype, including wt, homo, and hetero mice aged 8 and 30 days.

From each group (at least five animals per group), spleens from single mice were placed in Petri dishes containing Roosevelt Memorial Park Institute (RPMI) 1640 (Milteny Biotec, Bergisch Gladbach, GE) plus streptomycin (100 mg/mL) (Biowhittaker, Walkersville, USA) and 1% penicillin (Biowhittaker), and passed through a cell strainer with a 70 μm nylon membrane (Becton Dickinson, Bedford, MA). Afterwards, the homogenates were resuspended in RPMI 1640 containing 1 mM sodium pyruvate (Sigma-Aldrich, St. Louis, MO., USA), 1 M Hepes buffer (Sigma), gentamicin (10 mg/mL), streptomicin (10 ng/mL]-penicillin (100 UI)) (BioWhittaker, Walkersville, USA), and 30% fetal bovine serum (GIBCO, Carlsbad, CA, USA). Cell suspensions were counted using trypan blue dye exclusion test, and cell density was adjusted at 1 × 10^6^/mL. One milliliter of spleen cell suspensions from single mice was put in plate culture. Then, cells were stimulated with phorbol myristate acetate (PMA) (Sigma) plus ionomycin (Sigma) and left to incubate for 24 h at 37 °C in a 5% CO_2_ atmosphere. Harvested cells were dispensed into microeppendorf cups and centrifuged at 10,000× *g* for 10 min at 4 °C. Supernatants were collected and stored at −30 °C until use.

### 4.4. Cytokine Evaluation by Flow Cytometry

Cell supernatants obtained from single mice were incubated with the cytometric bead array (CBA) mouse soluble protein Flex Set kit (Becton Dickinson, Milan, Italy). This permitted to quantify the following cytokines: IL-12; IL-2; IFN-γ; IL-4; IL-10; IL-17; IL-6; and TNF-α. The capture beads provided in each CBA mouse soluble protein Flex Set were dispensed in each cell supernatant and, after 1 h incubation in the dark, a mouse soluble protein Flex Set PE detection was added for 2 h in the dark. Afterwards, each sample was suspended in 1 mL PBS and centrifuged at 800× *g* for 10 min. Then supernatants were carefully discarded and, finally, pellets were resuspended in 500 µL of PBS. Samples were acquired using the FACS DIVA software (Becton Dickinson), which allows distinguishing different types of cytokines according to some properties, such as forward scatter, side scatter, and mean fluorescence of the following fluorochromes: PE, APC, and APC-Cy7. A standard curve for all cytokines was initially prepared, and the FCAP software (Becton Dickinson) was then used to quantify all cytokines in standards and samples.

### 4.5. Statistical Analysis

Statistical analysis was performed using Bonferroni’s test for comparison between groups. GraphPad Prism statistical software release 5.0 for Windows Vista was used. The statistical significance was set at *p* < 0.05.

## Figures and Tables

**Figure 1 molecules-24-02205-f001:**
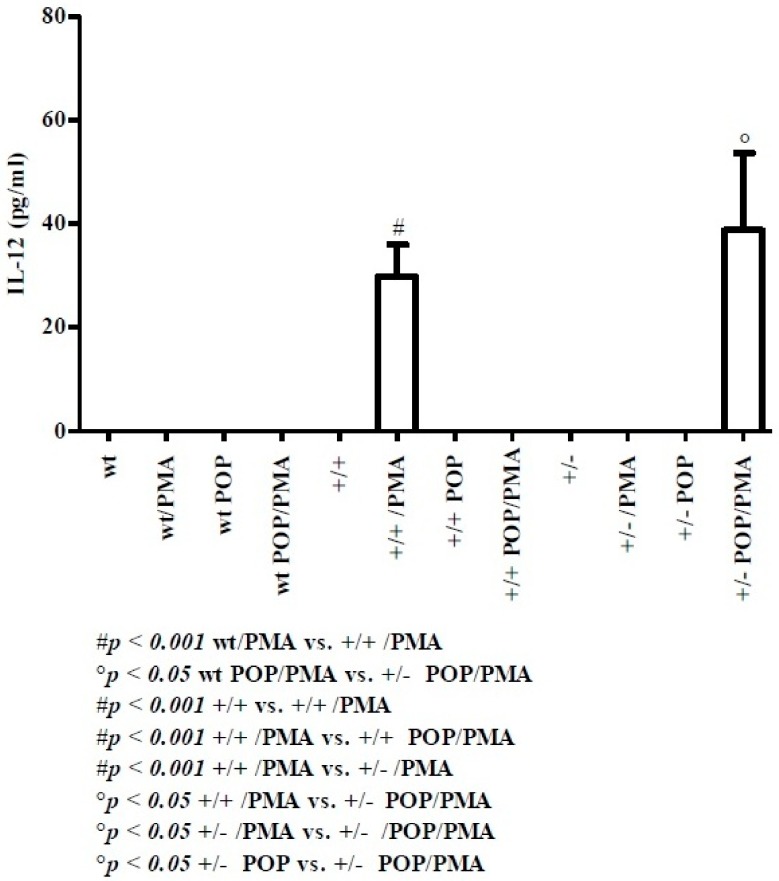
Levels of IL-12 in splenic supernatants from cell cultures of postnatal day 8 pups. Abbreviations: wt = wild type; +/+ = homo, +/− = hetero; PMA: phorbol myristate acetate, POP: suckling pups that received milk from dams administered with polyphenols. Evaluation of cytokines was performed using a cytometric bead array (CBA) mouse soluble protein Flex Set kit, as described in the Materials and Methods section. Statistical analysis was performed using Bonferroni’s test for comparison between groups. GraphPad Prism statistical software release 5.0 for Windows Vista was used. Statistical significance was set at *p* < 0.05.

**Figure 2 molecules-24-02205-f002:**
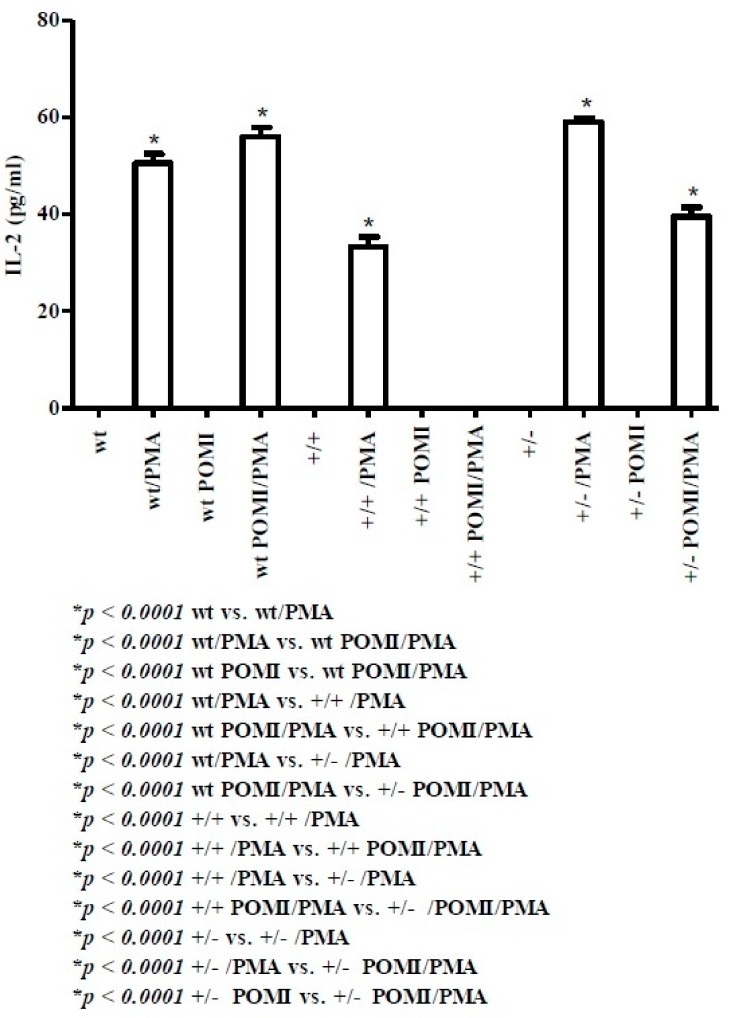
Levels of IL-2 in splenic supernatants from cell cultures of mice at day 30. Abbreviations: wt = wild type; +/+ = homo, +/− = hetero; POMI: post-weaning mice that received conventional feed supplemented with polyphenols. For methods and statistical analysis, see Figure 1.

**Figure 3 molecules-24-02205-f003:**
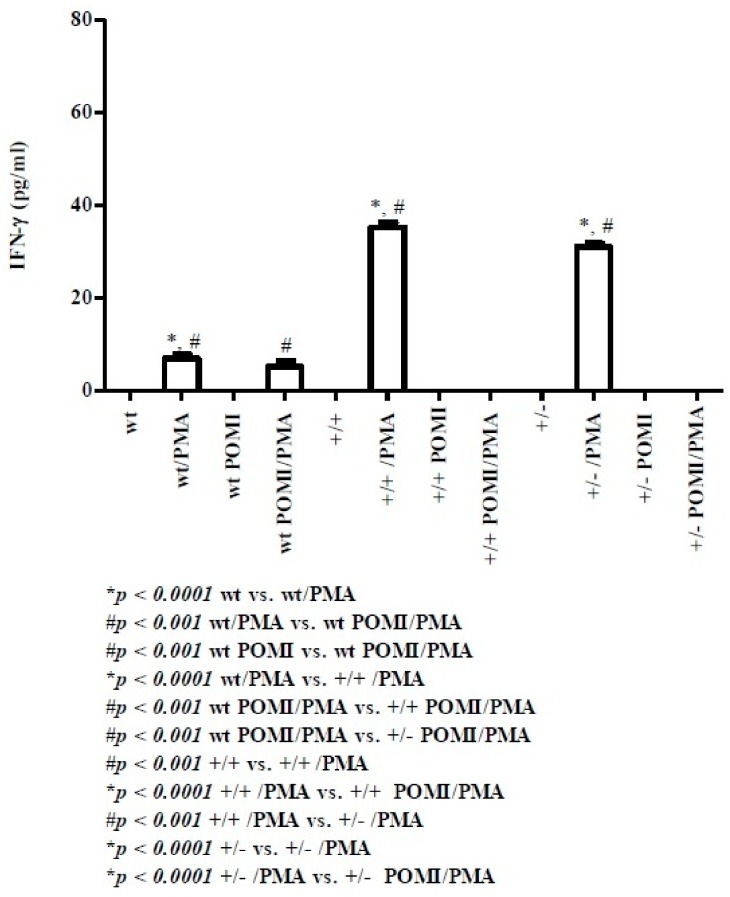
Levels of IFN-γ in splenic supernatants from cell cultures of mice at day 30. For methods, abbreviations, and statistical analysis, see Figure 1 and Figure 2.

**Figure 4 molecules-24-02205-f004:**
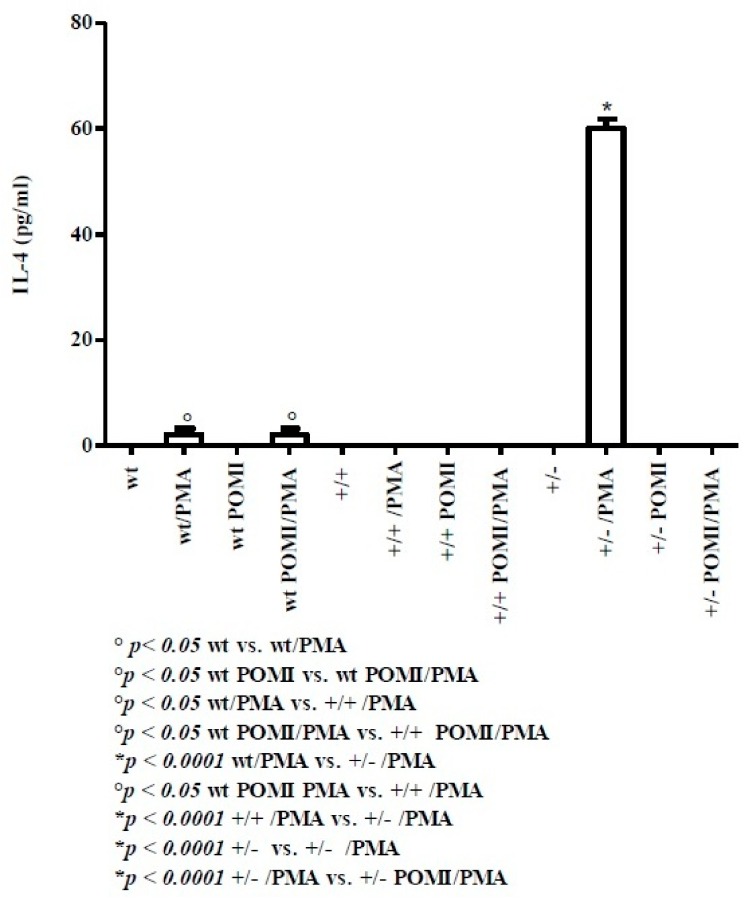
Levels of IL-4 in splenic supernatants from cell cultures of mice at day 30. For methods, abbreviations, and statistical analysis, see Figure 1 and Figure 2.

**Figure 5 molecules-24-02205-f005:**
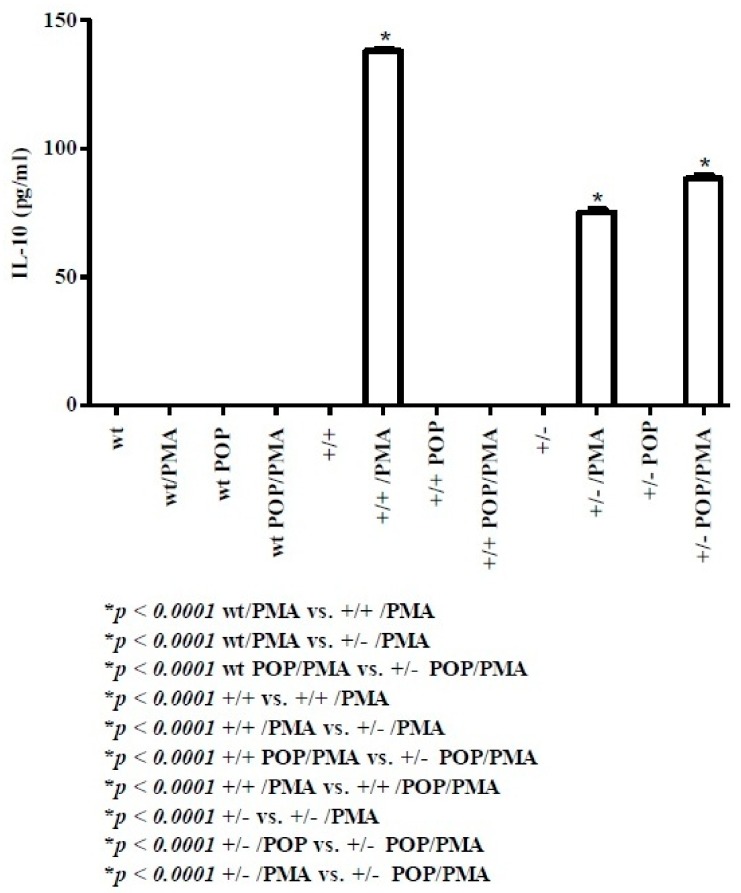
Levels of IL-10 in splenic supernatants from cell cultures of postnatal day 8 pups. For methods, abbreviations, and statistical analysis, see Figure 1 and Figure 2

**Figure 6 molecules-24-02205-f006:**
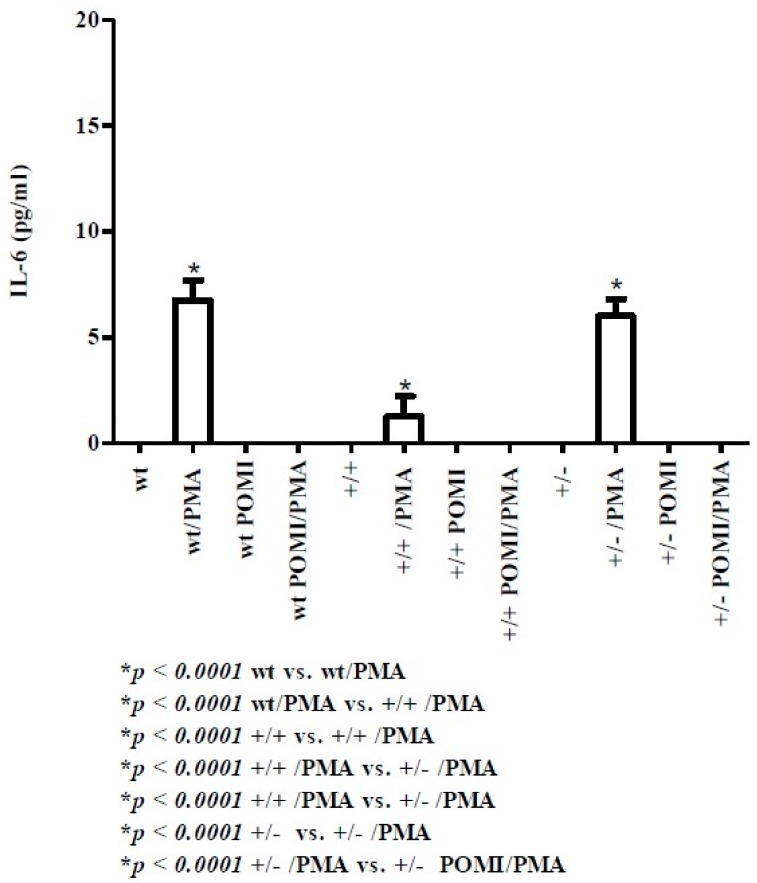
Levels of IL-6 in splenic supernatants from cell cultures of mice at day 30. For methods, abbreviations, and statistical analysis, see Figure 1 and Figure 2.

**Figure 7 molecules-24-02205-f007:**
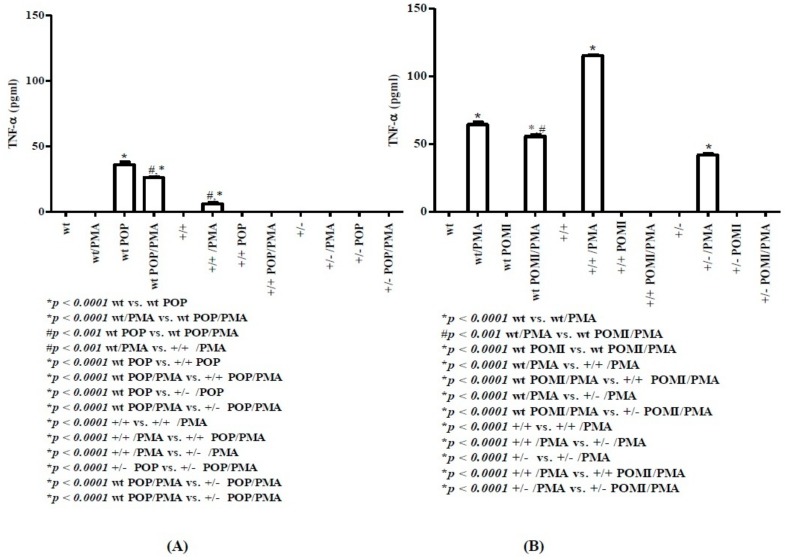
Levels of TNF-α in splenic supernatants from cell cultures of mice at day 8 (**A**) and day 30 (**B**), respectively. For methods, abbreviations, and statistical analysis, see Figure 1 and Figure 2.

**Figure 8 molecules-24-02205-f008:**
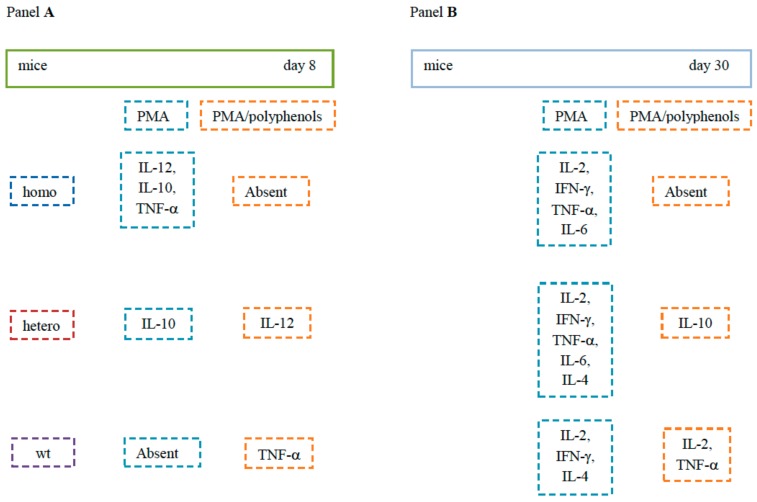
Time of appearance of splenic cytokines and polyphenol effects. In **A**, at day 8, cytokine release by PMA and PMA/polyphenol-stimulated splenocytes, respectively, is indicated. In homo mice, polyphenols abrogate release of IL-12, IL-10 and TNF-α. In hetero and wt mice, effects are less evident in view of a likely delayed maturation of immune cells. In **B**, at day 30, cytokine release by PMA and PMA/polyphenol-stimulated splenocytes, respectively, is indicated. Maturation of immune cells is more marked in homo and hetero mice than in wt mice. Polyphenols dampen the release of regulatory and inflammatory cytokines in homo and hetero mice and to a lesser extent in wt mice.

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
