# Peer review of "Polyphenol Effects on Splenic Cytokine Response in Post-Weaning Contactin 1-Overexpressing Transgenic Mice"

_molecules, 2019, doi:10.3390/molecules24122205_

Reviewer 1 Report

The Manuscript „Polyphenol effects on splenic cytokine response in post weaning Contactin 1-overexpressing transgenic mice” is interesting and generally well-written. Authors should, however, consider the following issues:

·  Introduction section should be shortened and the aim of the study ought to be more emphasized. In my opinion, within the Introduction, the Authors focus too much on the results of other experiments and describe them in great detail.

·  The Authors refer to the “companion paper (manuscript in reviewing)” [ln  469-477]. The Editor should decide if it is compatible with general publishing standards. In my opinion this fragment is not necessary and may be removed.

·  The manuscript should be thoroughly reviewed for editorial errors – e.g. line 92; 129; 411; 538.

Author Response

The Manuscript „Polyphenol effects on splenic cytokine response in post weaning Contactin 1-overexpressing transgenic mice” is interesting and generally well-written. Authors should, however, consider the following issues:

 Introduction section should be shortened and the aim of the study ought to be more emphasized. In my opinion, within the Introduction, the Authors focus too much on the results of other experiments and describe them in great detail.

R: Introduction has been curtailed according to referee suggestion (lines 70-71).

The Authors refer to the “companion paper (manuscript in reviewing)” [ln  469-477]. The Editor should decide if it is compatible with general publishing standards. In my opinion this fragment is not necessary and may be removed.

R: The sentences related to the companion paper have been deleted in the text.

The manuscript should be thoroughly reviewed for editorial errors – e.g. line 92; 129; 411; 538.

R: Errors and imperfections have been amended throughout the text.

Reviewer 2 Report

The authors analyzed cytokine production profile of wild-type animals and mice homozygous and heterozygous for the TAG/F3 mutation in the presence and absence of polyphenol-enriched diet. Contactin-overexpressing TAG/F3 mice, in general, exhibited a higher cytokine production induced by PMA+ionomycin treatment, which was abrogated by polyphenols. While these findings are significant and novel, the scientific merit of the publication is limited by the fact that it reports the results of one single experiment, as far as I can tell (five animals selected from each experimental group, each analyzed by the multiplex bead array Flex Set). This shortcoming must be eliminated, i.e. the experiment must be repeated in order to establish the reproducibility of the results. Other comments:

1. The abstract is misleading, difficult to understand in light of the title of the manuscript. The title suggests that polyphenol effects were examined, and these are not even mentioned in the results section of the abstract. The abstract must be rewritten so as to correspond to the title.

2. Regarding the sentence “Suckling pups from polyphenol-administered dams as well as from day 30 post weaning pups, …, have been used as animal models” in the abstract. The sentence suggests that the second group of animals contains suckling pups from post-weaning pups, which does not make sense.

3. The abbreviation “POMI” must be interpreted upon first mention on page 5, since the method section of the paper is at the end.

4. On p14 it is stated that cells were centrifuged at 10,000 RPM in order to separate the supernatant from the pellet. This must roughly correspond to 10,000 g, which is an unusually high speed for cells. Please provide centrifugation speed in RCF (relative centrifugal force), and show that this high speed does not affect cell viability.

5. Contactin is expressed in the nervous system, therefore differences between the cytokine production profile of isolated splenocytes of w.t. and contactin-overexpressing mice are not trivial to explain. While the authors refer to some references at the bottom of p9, they slur over the fact whether the changes, reported in refs 30-32 and 44, are present only in the nervous system. The authors must try to suggest possible explanations for their findings, explicitly considering the fact that contacting expression is limited to the nervous system.

Author Response

The authors analyzed cytokine production profile of wild-type animals and mice homozygous and heterozygous for the TAG/F3 mutation in the presence and absence of polyphenol-enriched diet. Contactin-overexpressing TAG/F3 mice, in general, exhibited a higher cytokine production induced by PMA+ionomycin treatment, which was abrogated by polyphenols. While these findings are significant and novel, the scientific merit of the publication is limited by the fact that it reports the results of one single experiment, as far as I can tell (five animals selected from each experimental group, each analyzed by the multiplex bead array Flex Set). This shortcoming must be eliminated, i.e. the experiment must be repeated in order to establish the reproducibility of the results. Other comments:

R: In the Materials and Methods section, it has been clarified that cytokine evaluation was performed on supernatants of cell cultures from single individuals from at least five mice per each group (lines 507-508; line 516; line 522).

The abstract is misleading, difficult to understand in light of the title of the manuscript. The title suggests that polyphenol effects were examined, and these are not even mentioned in the results section of the abstract. The abstract must be rewritten so as to correspond to the title.

R: The Abstract has been integrated with the results related to polyphenol effects (lines 24-28).

Regarding the sentence “Suckling pups from polyphenol-administered dams as well as from day 30 post weaning pups, …, have been used as animal models” in the abstract. The sentence suggests that the second group of animals contains suckling pups from post-weaning pups, which does not make sense.

R: The sentence has been corrected accordingly.

The abbreviation “POMI” must be interpreted upon first mention on page 5, since the method section of the paper is at the end.

R: It was done

On p14 it is stated that cells were centrifuged at 10,000 RPM in order to separate the supernatant from the pellet. This must roughly correspond to 10,000 g, which is an unusually high speed for cells. Please provide centrifugation speed in RCF (relative centrifugal force), and show that this high speed does not affect cell viability.

R: It was done (line 519).

Contactin is expressed in the nervous system, therefore differences between the cytokine production profile of isolated splenocytes of w.t. and contactin-overexpressing mice are not trivial to explain. While the authors refer to some references at the bottom of p9, they slur over the fact whether the changes, reported in refs 30-32 and 44, are present only in the nervous system. The authors must try to suggest possible explanations for their findings, explicitly considering the fact that contacting expression is limited to the nervous system.

R: It was clarified in the text that further experiments are needed to establish the relationship between nervous system and immune system alterations in TAG/F3 mice (lines 409-411).

Round  2

Reviewer 2 Report

The authors' response is accepted.